# Antiprotozoal Activity Against *Entamoeba histolytica* of Flavonoids Isolated from *Lippia graveolens* Kunth

**DOI:** 10.3390/molecules25112464

**Published:** 2020-05-26

**Authors:** Ramiro Quintanilla-Licea, Javier Vargas-Villarreal, María Julia Verde-Star, Verónica Mayela Rivas-Galindo, Ángel David Torres-Hernández

**Affiliations:** 1Facultad de Ciencias Biológicas, Universidad Autónoma de Nuevo León (UANL), Av. Universidad S/N, Cd. Universitaria, San Nicolás de los Garza, C.P. 66455 Nuevo León, Mexico; maria.verdest@uanl.edu.mx (M.J.V.-S.); angel.torreshr@uanl.edu.mx (Á.D.T.-H.); 2Laboratorio de Bioquímica y Biología Celular, Centro de Investigaciones Biomédicas del Noreste (CIBIN), Dos de abril esquina con San Luis Potosí, C.P. 64720 Monterrey, Mexico; jvargas147@yahoo.com.mx; 3Facultad de Medicina, Universidad Autónoma de Nuevo León (UANL), Madero y Aguirre Pequeño, Mitras Centro, Monterrey, C.P. 64460 Nuevo León, Mexico; veronica.rivasgl@uanl.edu.mx

**Keywords:** infectious diseases, amoebiasis, Mexican oregano, bioguided isolation, flavonoids, antiprotozoal agents

## Abstract

Amebiasis caused by *Entamoeba histolytica* is nowadays a serious public health problem worldwide, especially in developing countries. Annually, up to 100,000 deaths occur across the world. Due to the resistance that pathogenic protozoa exhibit against commercial antiprotozoal drugs, a growing emphasis has been placed on plants used in traditional medicine to discover new antiparasitics. Previously, we reported the in vitro antiamoebic activity of a methanolic extract of *Lippia graveolens* Kunth (Mexican oregano). In this study, we outline the isolation and structure elucidation of antiamoebic compounds occurring in this plant. The subsequent work-up of this methanol extract by bioguided isolation using several chromatographic techniques yielded the flavonoids pinocembrin (**1**), sakuranetin (**2**), cirsimaritin (**3**), and naringenin (**4**). Structural elucidation of the isolated compounds was achieved by spectroscopic/spectrometric analyses and comparing literature data. These compounds revealed significant antiprotozoal activity against *E. histolytica* trophozoites using in vitro tests, showing a 50% inhibitory concentration (IC_50_) ranging from 28 to 154 µg/mL. Amebicide activity of sakuranetin and cirsimaritin is reported for the first time in this study. These research data may help to corroborate the use of this plant in traditional Mexican medicine for the treatment of dyspepsia.

## 1. Introduction

Amoebiasis is caused by *Entamoeba histolytica*, which is a protozoan of the family Endomoebidae [1]. It is related to elevated morbidity and mortality worldwide, and has become a serious public health problem in developing countries [2]. Traveling to endemic countries is a risk factor for acquiring an *E. histolytica* infection [3]. After malaria, amoebiasis is the second cause of death due to parasitic diseases [4,5]. The symptoms vary from mild diarrhea to dysentery, but, occasionally, *E. histolytica* can invade the intestinal mucosal barrier and trigger liver abscesses [6]. Asymptomatic infections occur in 90% of individuals, whereas the remaining 10% contract symptomatic infections [7]. Around 50 million people suffer from severe amoebiasis, and 40,000–100,000 deaths occur annually due to this parasitosis [8,9].

Currently, metronidazole is the most used commercial drug for the treatment of amoebiasis, however, since drug resistance by *E. histolytica* is increasing, the use of higher doses to overcome the infection is needed, thus causing unpleasant side effects [10,11]. Considering these undesired side effects as well as the development of resistant strains of *E. histolytica* against metronidazole, more efficient and safer antiamoebic agents are required [12,13,14].

Natural products occurring in medicinal plants have proved to be an important source of leading compounds for the design of new drugs [15]. Mexican oregano (*Lippia graveolens* Kunth) has been used in traditional Mexican medicine for curing inflammation-related diseases, such as respiratory and digestive disorders, headaches, and rheumatism, among others [16,17]. Oregano’s essential oil, regardless of the species, shows a broad range of effects on bacteria, with some of them being resistant to antibiotics of clinical use, as well as on fungi and parasites [18,19,20,21,22].

Recently, we reported the in vitro antiamoebic activity of a methanolic extract of *Lippia graveolens* Kunth and the bioguided isolation of carvacrol, as one of the bioactive compounds with antiprotozoal activity [23]. This work aims to isolate and achieve the structure elucidation of additional antiamoebic compounds present in this plant.

## 2. Results

### 2.1. Bioguided Isolation of Flavonoids from Lippia graveolens Kunth

As previously reported, the partition of a methanolic extract of *Lippia graveolens* by extraction with *n*-hexane and fractionation of the hexane phase led to the isolation of carvacrol with excellent antiamoebic activity [23]. The subsequent handling of the remaining methanol (MeOH) by partition between methanol-water and ethyl acetate (EtOAc), carried out in this research, yielded an EtOAc residue with 93.3% growth inhibition of *E. histolytica*. After column chromatography (silica gel, Sephadex), this residue afforded the known flavonoids pinocembrin (**1**), sakuranetin (**2**), cirsimaritin (**3**), and naringenin (**4**), with significant antiamoebic activity (Figure 1).

The isolated flavonoids were identified by comparing their physical and spectral data with those reported in the literature.

The electron ionization (EI) mass spectrum showed a molecular ion with *m*/*z* = 256 for pinocembrin **1** (calcd. for C_15_H_12_O_4_, 256.253), *m*/*z* = 286 for sakuranetin **2** (calcd. for C_16_H_14_O_5_, 286.283), *m*/*z* = 314 for cirsimaritin **3** (calcd. for C_17_H_14_O_6_, 314.28), and *m*/*z* = 272 for naringenin **4** (calcd. for C_15_H_12_O_5_, 272.252).

The infrared (IR) spectrum of all isolated flavonoids (see Appendix A) contained absorption bands at 1600–1650 cm^−1^ (medium) and 1100–1250 cm^−1^ (strong), consistent with a C=O bond in the molecules [24,25,26,27].

One- and two-dimensional nuclear magnetic resonance (NMR) spectra were recorded for the isolated compounds using deuterated dimethyl sulfoxide (DMSO-*d*_6_). ^1^H- and ^13^C-NMR chemical shifts (see Appendix A) were in accordance with those reported for pinocembrin **1** [28,29,30], sakuranetin **2** [31,32,33,34], cirsimaritin **3** [26,35,36,37], and naringenin **4** [27,28,38,39,40,41]. An unambiguous assignment of the ^13^C-NMR spectrum of these compounds was deduced from ^1^H-^1^H COSY, NOESY, HSQC, and HMBC spectra (see Appendix A).

The four isolated flavonoids have the common characteristic that the rotameric hydroxy group at C-5 forms an intramolecular H-bond with the carbonyl group [42]. That explains the shift of this proton absorption to the range of about δ 13.0–12.0 as a sharp singlet in DMSO-*d*_6_ [43].

### 2.2. Entamoeba Histolytica Growth Parameters

After collecting data from *E. histolytica* growth kinetics experiments, it was estimated that the generation time is 14.76 h and the duplication time is 21.30 h.

The ideal growth time of *E. histolytica* for evaluating the amebicide activity of the compounds was set at 72 h because, at this point, protozoa are still in the exponential and sustained growth stage, thus decreasing the number of false positives.

### 2.3. In Vitro Assay for Entamoeba histolytica

The pure compounds were dissolved in DMSO to a concentration of 150 µg/mL in a suspension of *E. histolytica* trophozoites in a logarithmic phase in peptone, pancreas, and liver extract plus 10% bovine serum (PEHPS medium). They showed significant growth inhibition of *E. histolytica* at this concentration. The 50% inhibitory concentration (IC_50_) values of these compounds ranged from 28.86 to 154.26 µg/mL (metronidazole IC_50_ 0.205 µg/mL).

Figure 2, Figure 3, Figure 4 and Figure 5 show the 50% inhibitory concentration of each compound calculated by using a Probit analysis, considering a 95% confidence level.

## 3. Discussion

It is estimated that about 6000 flavonoids are present in different plants worldwide [44,45], and many of them are common ingredients of our daily food. Flavonoids exhibit a variety of biological properties, such as antioxidant, anticancer, antibacterial, antifungal, antiparasitic [46,47,48,49,50,51,52,53], as well as those for treating other kinds of illness [54,55,56,57].

More than 20 flavonoids have been identified in the leaves of *Lippia graveolens* by high pressure liquid chromatography (HPLC), according to previous reports [58]. Pinocembrin, sakuranetin, naringenin, and cirsimaritin have already been isolated from this plant [58,59,60,61,62], but, in the respective studies, no antiamoebic activity is reported. Nevertheless, some research groups have reported antiprotozoal activity of these flavonoids isolated from sources other than *Lippia graveolens.*

A weak antiparasitic activity of pinocembrin has been shown against trypomastigotes of *Trypanosoma cruzi*, with inhibition values in the range of 40.13% (at 250 µg/mL) to 43.68% (at 500 µg/mL), according to Grael et al. [63]. Weak inhibition was also observed against *Giardia lamblia* trophozoites, with an IC_50_ of 174.4 µg/mL reported by Alday-Provencio et al. [64] and an IC_50_ of 57.39 µg/mL reported by Calzada et al. [65], respectively.

Sakuranetin presented activity against *Leishmania amazonensis*, *Leishmania brazilians*, *Leishmania major*, and *Leishmania chagasi*, with a range of 43–52 µg/mL, as well as against *T. cruzi* trypomastigotes, with an IC_50_ of 20.17 µg/mL, according to Grecco et al. [39].

Regarding parasitic diseases, cirsimaritin showed potent inhibition against *Plasmodium falciparum* resistant to chloroquine, with an IC_50_ of 16.9 µM [66], and similar activity against *Leishmania donovani* (IC_50_ = 3.9 µg/mL), *Trypanosoma brucei rhodesiense* (IC_50_ = 3.3 µg/mL), and *T. cruzi* (IC_50_ = 19.7 µg/mL), according to Tasdemir et al. [49].

Antiparasitic research has found giardicidal activity in naringenin (**4**), with an IC_50_ of 125.7 µg/mL [64] and 47.84 µg/mL [65], according to Alday-Provencio et al. and Calzada et al., respectively, but no damage against *Trypanosoma cruzi* and *Leishmania* spp. was observed, according to Grecco et al. [39].

Some results of pinocembrin and naringenin tested against the strain *E. histolytica* HM1:IMSS have been published; pinocembrin isolated from *Teloxys graveolens* and naringenin from a commercial source exhibited an IC_50_ of 80.76 and 98.24 µg/mL, respectively [65,67]. This indicates a lower effectiveness compared with our results (IC_50_ of 29.51 µg/mL for pinocembrin and 28.85 µg/mL for naringenin, respectively), which could be explained by the difference between the methodology reported by Calzada [65] and that used by our group. The principal difference was the time that the trophozoites were exposed to the chemical compound. In our methodology, we incubated the trophozoites with the flavonoids for 72 h, while Calzada [65] incubated them for 48 h, which could represent a factor in the certainty of the biological activity.

In our view, this is the first report on the amebicide activity of sakuranetin and cirsimaritin.

The antiprotozoal activity of pinocembrin and naringenin (IC_50_ of 29.63 and 28.86 µg/mL, respectively) was higher compared with sakuranetin (44.51 µg/mL), and the most remarkable comparison was with cirsimaritin (154.26 µg/mL), revealing that a 5,7-dihydroxylated A ring is essential for antiprotozoal activity, as remarked by Calzada [65]. The structure–effect correlations also showed that a 2,3-double bond in ring B (as in cirsimaritin) reduces the antiprotozoal activity.

Currently, there are no available studies on the mechanism of action against *E. histolytica* of the flavonoids isolated from *L. graveolens*, although there are reports on some structurally related flavonoids [68,69]. The ultrastructural changes in the morphology of *Entamoeba histolytica* when it was assayed with (−)-epicatechin, a flavan-3-ol flavonoid, have previously been demonstrated, showing an IC_50_ of 1.9 µg/mL. The results indicated programmed cell death activation with nuclear alterations (small clumps around the nuclear membrane), in addition to cytoplasmatic modifications, such as an increase of glycogen deposits and a reduction of the size and number of vacuoles [70]. Recently, Bolaños et al. [68] have also demonstrated that the flavonoids (−)-epicatechin and kaempferol affect cytoskeleton proteins and functions in *E. histolytica* [68,71], leading to changes in essential cellular mechanisms, such as adhesion, migration, phagocytosis, and cytolysis. These findings lead us to have an idea of the possible targets and mechanisms of action of the flavonoids isolated from *L. graveolens*. None of the flavonoids isolated from *L. graveolens* present a hydroxyl group at position 3, as in the case of epicatechin and kaempferol, so there must be subtle differences in the mechanism of action of pinocembrin, sakuranetin, naringenin, and cirsimaritin, and part of the future work of our group will be devoted to this issue.

The significant inhibitory effect against *E. histolytica* observed for the methanolic extract of *Lippia graveolens* (IC_50_ = 59.15 µg/mL) can be attributed to the presence of the flavonoids isolated from the ethyl acetate partition as well as the carvacrol mainly obtained from the hexane partition [23]. Compounds with a higher polarity occurring in *L. graveolens*, soluble in methanol or water, are not involved in the antiprotozoal activity of this plant against *E. histolytica*.

## 4. Materials and Methods

### 4.1. General

An Electrothermal 9100 apparatus (Electrothermal Engineering Ltd., Southend-on-Sea, UK) was used for melting point acquisition. IR spectra were measured on a Frontier Fourier transform infrared (FT-IR) spectrometer (PerkinElmer, Waltham, MA, USA) with an ATR accessory. NMR spectra were recorded on an Avance DPX 400 spectrometer (Bruker, Billerica, MA, USA) running at 400.13 MHz for ^1^H and 100.61 MHz for ^13^C. EI-MS were obtained on a MAT 95 spectrometer (70 eV, Finnigan, San Jose, CA, USA). Thin layer chromatography (TLC) was realized on precoated silica gel glass plates (5 × 10 cm, Merck silica gel 60 F_254_, Darmstadt, Germany). Column chromatography was carried out on silica gel (60–200 mesh) purchased from J. T. Baker (Phillipsburg, NJ, USA). Size-exclusion chromatography was performed on Sephadex LH-20 (Lipophilic Sephadex, Amersham Biosciences Ltd., purchased from Sigma-Aldrich Chemie, Steinheim, Germany).

### 4.2. Plant Material

Aerial parts of *Lippia graveolens* were collected near the town General Cepeda (Mexican State Coahuila) in March 2011 and identified by Maria del Consuelo González. A voucher specimen (No. 025554) was deposited at the Herbario de la Facultad de Ciencias Biológicas (UANL), Nuevo León, México. The plant name has been checked with http://www.theplantlist.org. The vegetal material was dried and ground to powder.

### 4.3. Plant Extraction and Bioguided Isolation of Antiamoebic Compounds from Lippia graveolens Kunth

In total, 600 g of dried and ground *Lippia graveolens* leaves were extracted in a Soxhlet apparatus for 40 h with MeOH. After filtration, the solvent was removed in a rotatory evaporator to yield 260 g of crude extract. This extract was analyzed for its amebicide activity on trophozoites of *E. histolytica* (HM1:IMSS strain), showing a significant inhibition percentage (89%; IC_50_ 59.15 µg/mL) in terms of the standard concentration of 150 µg/mL. Afterward, the extract was redissolved in 2 L methanol and divided into four portions of 500 mL each for conducting liquid-liquid partition with *n*-hexane. After solvent evaporation, 19.9 g of a residue with high amebicidal activity (90.9% growth inhibition) was obtained. Bioguided fractionation of this hexane partition, using column chromatography on silica gel, provided 2.2 g of carvacrol (98.4% growth inhibition; IC_50_ 44.30 µg/mL), as previously reported [23].

The methanolic phase was concentrated under reduced pressure up to a volume of ca. 500 mL, and 1.5 L distilled water was then added, gradually and under constant stirring. Afterward, the methanol/water mixture was divided into four portions of 500 mL each and submitted to liquid–liquid partition with ethyl acetate to yield, after solvent evaporation, 67.3 g of a combined residue with 93.3% growth inhibition against *E. histolytica*.

The EtOAc partition was suspended in 400 mL of chloroform (CHCl_3_), and, after stirring, filtration, and solvent evaporation, 25.8 g of CHCl_3_-soluble residue was obtained. The material recovered from the filter was then suspended in 400 mL of EtOAc, and, after stirring, filtration, and solvent evaporation, 8.8 g of EtOAc-soluble residue was obtained. This second material recovered from the filter was then suspended in 100 mL MeOH, and the same procedure was applied to yield 8.8 g of MeOH-soluble residue and ca. 24 g of an insoluble powder from the final filtration. Each residue was analyzed for its anti-*Entamoeba histolytica* activity, and remarkable results were obtained, mainly for the CHCl_3_-residue, with 90.9% growth inhibition, followed by the EtOAc residue, with 61.6% growth inhibition. The MeOH residue and insoluble powder did not show any activity.

The CHCl_3_ fraction was divided into five portions of ca. 5 g, and each of them was chromatographed on a silica gel (100 g) column (60 × 2.6 cm) and eluted with stepwise gradients of chloroform-ethyl acetate, and finally with methanol. For each column, a total of 110 subfractions (50 mL) were obtained and collected, and considering their TLC (CHCl_3_–EtOAc, 9:1) profiles, split into eight main fractions (A-H). These main fractions, containing the nonpolar to the more polar compounds, were used for amebicide assays. Out of the eight fractions, only fractions C, D, E, and F showed amebicide activity, with 95.34%, 96.89%, 97.24%, and 95.85% growth inhibition, respectively.

Fraction C (showing the main compound, according to TLC, with *R_f_* = 0.77; CHCl_3_-Ethyl acetate, 9:1) was divided into five portions of approx. 1 g, and each of them was chromatographed on a silica gel (20 g) column (39 × 2 cm) and eluted with a stepwise gradient solvent system of chloroform and ethyl acetate, and finally with methanol. Sixty subfractions (10 mL) were collected for each column and combined, based on their TLC (CHCl_3_-Ethyl acetate, 9:1) profiles, into five main fractions (CA-CE). Fraction CB presented the main compound, with *R_f_* = 0.77, so it was subjected to subsequent purification with three columns packed with silica gel as stationary phase (data not shown). Afterward, 315 mg of a viscous liquid with a high amebicide activity (96.76% growth inhibition; IC_50_ 44.1 µg/mL) was recovered. The spectroscopic results indicated that this compound is carvacrol, previously isolated from hexane partition [23].

Fraction D (showing the main compound, according to TLC, with *R_f_* = 0.51; CHCl_3_-Ethyl acetate, 9:1) was divided into four portions of ca. 1 g and each of them was submitted to chromatography on a silica gel (20 g) column (39 × 2 cm) and eluted with a stepwise gradient solvent system of chloroform and ethyl acetate, and finally with methanol. Sixty subfractions (10 mL) were collected for each column and combined, based on their TLC (CHCl_3_-Ethyl acetate, 9:1) profiles, into five main fractions (DA-DE). Fraction DB presented the main compound, with *R_f_* = 0.51, so it was subjected to subsequent purification with several columns packed with Sephadex as stationary phase (data not shown). Afterward, 32 mg of a solid with a high amebicide activity (89.63% growth inhibition; IC_50_ 29.63 µg/mL) was recovered. The spectroscopic results indicated that this compound is pinocembrin **1** (C_15_H_12_O_4_; M.p. 210 °C: Lit. 223–236 °C [72], 191–193 °C [24]).

Fraction E (showing the main compound, according to TLC, with *R_f_* = 0.33; CHCl_3_-Ethyl acetate, 9:1) was divided into three portions of approx. 1 g and each of them was chromatographed on a silica gel (20 g) column (39 × 2 cm) and eluted with a stepwise gradient solvent system of chloroform and ethyl acetate, and finally with methanol. Sixty subfractions (10 mL) were collected for each column and combined, based on their TLC (CHCl_3_-Ethyl acetate, 9:1) profiles, into five main fractions (EA-EE). Fractions EB and EC presented the main compound, with *R_f_* = 0.33, so they were subjected to subsequent purification with several columns packed with Sephadex as stationary phase (data not shown). Afterward, 102 mg of a solid with a high amebicide activity (97.24% growth inhibition; IC_50_ 44.51 µg/mL) was recovered. The spectroscopic results indicated that this compound is sakuranetin **2** (C_16_H_14_O_5_; M.p. 160 °C: Lit. 151–153 °C [73], 143–144 °C [74]).

Fraction F showed only a main compound, according to TLC, with *R_f_* = 0.10 (CHCl_3_-Ethyl acetate, 9:1). Using an eluent of a higher polarity (CHCl_3_-Ethyl acetate, 1:1), this compound was revealed to be a mixture of two compounds, with an *R_f_* of 0.70 and 0.53, respectively. This mixture was no longer soluble in chloroform and it was therefore not possible to use column chromatography with silica gel of a normal phase to try to separate the two compounds. Then, this fraction was suspended in 30 mL of chloroform and heated to reflux for 15 min, noting that a large quantity of the material remained insoluble. After cooling and filtering, 645 mg of an insoluble solid consisting of the two compounds of the mixture (TLC) was recovered, while in the chloroform solution, a mixture of the same compounds was also observed, but with additional impurities, mainly of a lower polarity. The insoluble solid (645 mg) recovered from the filter was suspended in 30 mL methanol and heated to reflux until complete solubility was observed. After cooling, the MeOH solution was kept refrigerated for 72 h with hermetic closure, and during that time, 50.6 mg of a greenish powder containing only the compound with *R_f_* = 0.53 (CHCl_3_-Ethyl acetate, 1:1) was separated. This solid presented no significant amebicide activity (52.45% growth inhibition; IC_50_ 154.26 µg/mL). The spectroscopic results indicated that this compound is cirsimaritin **3** (C_17_H_14_O_6_; M.p. 268 °C: Lit. 256–258 °C [26], 267–268 °C [75]).

The filtered MeOH solution was concentrated and chromatographed on a Sephadex (50 g) column (160 × 1.5 cm) and eluted with methanol (300 mL). A total of 60 subfractions (5 mL) were collected and combined, based on their TLC (CHCl_3_-Ethyl acetate, 1:1) profiles, into seven main fractions (FA-FG). From fraction FC, additional cirsimaritin with *R_f_* = 0.53 was recovered. Fraction FF showed only the pure compound, with *R_f_* = 0.70. This solid also presented high amebicide activity (96.50% growth inhibition; IC_50_ 28.86 µg/mL). The spectroscopic results indicated that this compound is naringenin **4** (C_15_H_12_O_5_; M.p. 254 °C: Lit. 250–252 °C [27], 248–250 °C [73]).

### 4.4. Antiprotozoal Assay

#### 4.4.1. Test Microorganisms

Strain HM-1:IMSS of *Entamoeba histolytica* was obtained from the microorganism culture collection of the Centro de Investigación Biomédica del Noreste (CIBIN) in Monterrey, Mexico. The trophozoites were grown axenically and maintained in peptone, pancreas, and liver extract plus bovine serum and employed at the log phase of growth (2 × 10^4^ cells/mL) by all of the bioassays performed [76,77].

The procedure for determining the growth curve for *E. histolytica* was performed in 13 × 100 mm screw cap test tubes, by inoculating 20,000 trophozoites of *E. histolytica* in 5 mL of PEHPS medium, to which 10% bovine serum was added. Subsequently, they were incubated at 36.5 °C for 120 h, and every 24 h, the number of trophozoites was determined and the growth parameters in the medium were evaluated. The process was conducted in three separate experiments per triplicate.

#### 4.4.2. In Vitro Assay for Entamoeba histolytica

Each compound was dissolved in DMSO and adjusted to a concentration of 150 µg/mL in a suspension of *E. histolytica* trophozoites at a logarithmic phase in PEHPS medium with 10% bovine serum. Vials were incubated for 72 h, and then chilled in iced water for 20 min, and, by using a hemocytometer, the number of dead trophozoites per milliliter was calculated. Each extract assay was carried out in triplicate. Metronidazole was used as a positive control, and as a negative control, an *E. histolytica* suspension in PEHPS medium with no extract added was used. The percentage of inhibition was estimated as the number of dead trophozoites compared to the negative controls.

#### 4.4.3. In Vitro IC_50_ Determination

Each compound was dissolved in dimethyl sulfoxide and adjusted to 150, 75, 37.5, 18.75, and 9.375 µg/mL by adding a suspension of *E. histolytica* trophozoites at a logarithmic phase in PEHPS medium with 10% bovine serum. Vials were incubated for 72 h, and then chilled in cold water for 20 min, and the number of dead trophozoites per milliliter was evaluated using a hemocytometer. All assays were performed in triplicate. Metronidazole was used as a positive control, and as a negative control, an *E. histolytica* suspension in PEHPS medium with no extract added was used. The percentage of inhibition was estimated as the number of dead trophozoites compared to the negative controls. The 50% inhibitory concentration of each compound was calculated by using a Probit analysis, considering a 95% confidence level.

## 5. Conclusions

The isolated and pure flavonoids from *L. graveolens* showed significant growth inhibition against *E. histolytica* (52% to 97% at a concentration of 150 µg/mL). The IC_50_ values of these compounds ranged from 28.86 to 154.26 µg/mL, so they were not as effective as metronidazole (IC_50_ 0.205 µg/mL), but these IC_50_ values can be used as a guideline for further research on this plant as a source of potential antiamoebic agents.

The main contribution of this research work lies in the fact that it has shown that the presence of the flavonoids described herein in *Lippia graveolens* has a direct relationship with the antiprotozoal activity of extracts of this plant against *Entamoeba histolytica*. These flavonoids could be used as biomarkers [78] for the quality control of phytotherapeutics developed based on this work.

The results of our research may also form the basis for directly incorporating the use of *Lippia graveolens* extracts into conventional and complementary medicine for the treatment of amebiasis, as well as other infectious diseases.

## Figures and Tables

**Figure 1 molecules-25-02464-f001:**
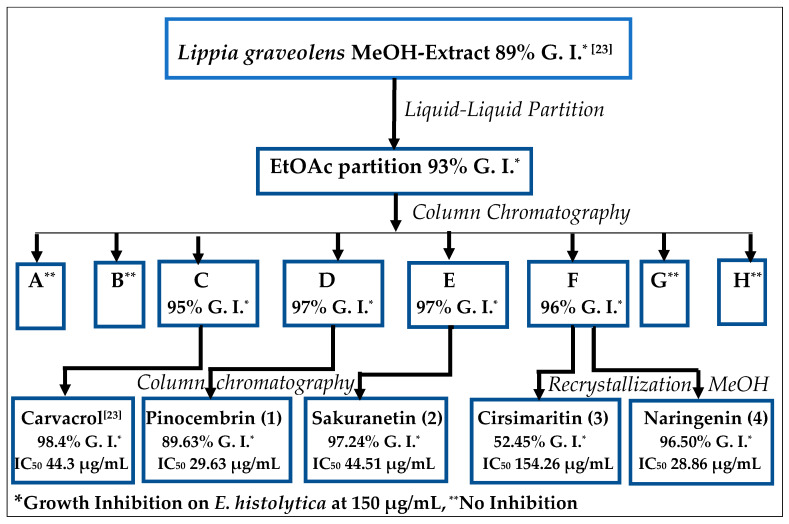
General scheme for the bioguided isolation of compounds with antiamoebic activity from *Lippia graveolens Kunth* (Mexican oregano).

**Figure 2 molecules-25-02464-f002:**
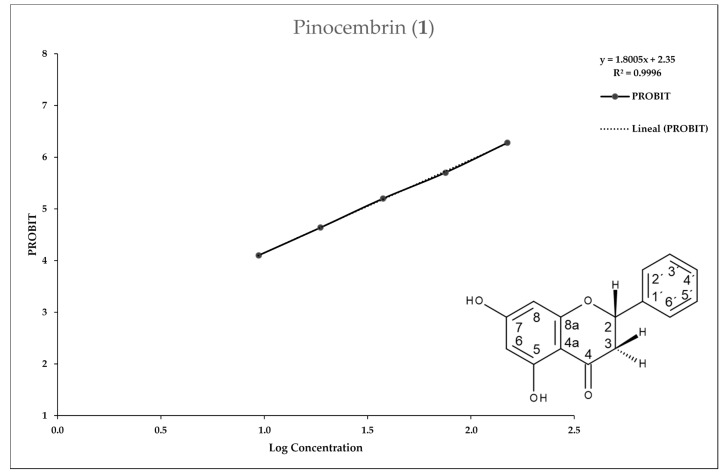
Antiprotozoal activity of Pinocembrin **1** against *Entamoeba histolytica*. Growth inhibition of 89.63% at 150 µg/mL, 50% inhibitory concentration value (IC_50_) = 29.63 µg/mL.

**Figure 3 molecules-25-02464-f003:**
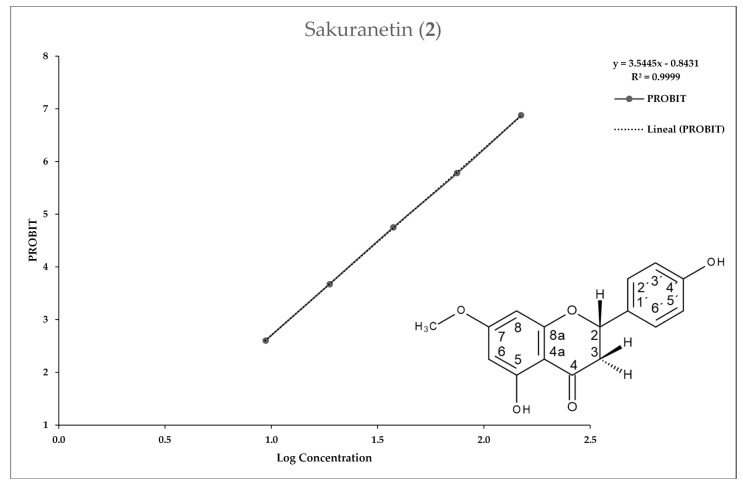
Antiprotozoal activity of Sakuranetin **2** against *Entamoeba histolytica.* Growth inhibition of 97.24% at 150 µg/mL, IC_50_ = 44.51 µg/mL.

**Figure 4 molecules-25-02464-f004:**
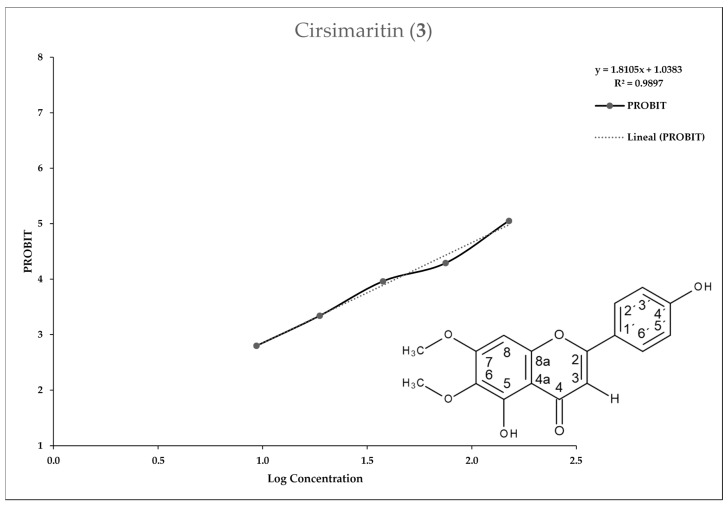
Antiprotozoal activity of Cirsimaritin **3** against *Entamoeba histolytica*. Growth inhibition of 52.45% at 150 µg/mL, IC_50_ = 154.26 µg/mL.

**Figure 5 molecules-25-02464-f005:**
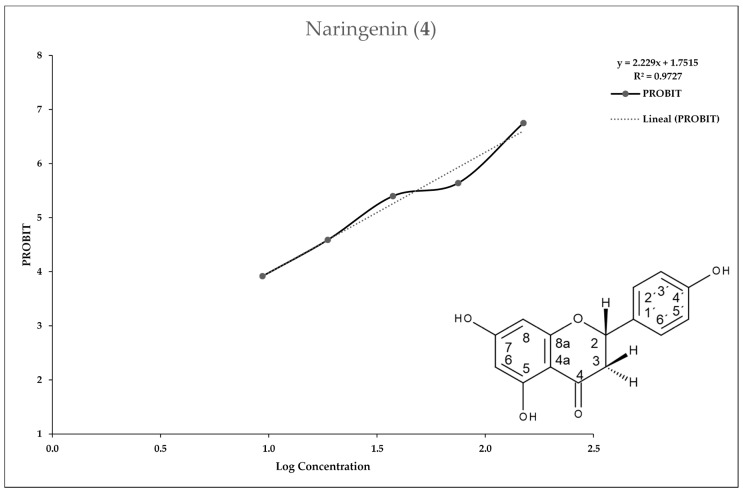
Antiprotozoal activity of Naringenin **4** against *Entamoeba histolytica*. Growth inhibition of 96.50% at 150 µg/mL, IC_50_ = 28.86 µg/mL.

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
