# Peer review of "Antiprotozoal Activity Against Entamoeba histolytica of Flavonoids Isolated from Lippia graveolens Kunth"

_molecules, 2020, doi:10.3390/molecules25112464_

Round 1

Reviewer 1 Report

I believe that this topic is not new since the identified compounds are known for antiprotozoal activity. Authors improve the manuscript however the topic is not new and in my opinion not suitable for publication in  Molecules

Author Response

Reviewer 1

Rev.

I believe that this topic is not new since the identified compounds are known for antiprotozoal activity.

Auth.

We know that flavonoids are distributed in a wide variety of plants, and sometimes the same flavonoid has been found in different species. However, it is necessary to carry out research such as ours where through bio-guided isolation and correct identification of the bioactive compounds, it is possible to assign pharmacological activity without any doubt as to the presence of these compounds in the plant. The main contribution of this research work lies in the fact that it has shown that the presence in Lippia graveolens of the flavonoids described herein has a direct relationship with the antiprotozoal activity of extracts of this plant. To our knowledge, this would be the first report for the amebicide activity of sakuranetin and cirsimaritin.

Reviewer 2 Report

All addressed remarks were answered

Author Response

Reviewer 2

Rev.

No comments or suggestions 

Auth.

Thank you for your kind evaluation.

Reviewer 3 Report

Reviewer #1:

Comments and Suggestions for Authors

The manuscript entitled: "Antiprotozoal Activity Against Entamoeba histolytica 2

 of Flavonoids Isolated from Lippia graveolens Kunth" done by Ramiro et al is an interesting work with good result of purification. However, there are several drawbacks should be carefully addressed before accepting this manuscript.

  1. Plagiarism is not acceptable in Moleculessubmissions, Plagiarism includes copying text, ideas, images, or data from another source, even from your own publications, without giving any credit to the original source. In the current manuscript 72% of plagiarism were detected through the manuscript by using turnitin (no containing the references section, I sent the file for authors checked). However, consideration the contribution of good data, I suggest that the current manuscript must be extensively modified and rewritten.
  2. Though the manuscript contains some novel points, but it seems being absent in the abstract. For example, “the amebicide activity of sakuranetin (2) and cirsimaritin (3) is new finding in this study”. All the novel findings should be highlighted in this section.
  3. The introduction should avoid secret the apart of result of the study (final paragraph, you tell the important result of your paper). This paragraph should mention the aims of the study, and the outline of the work.
  4. The data of bioactivity of all fractions, sub-fractions should be shown in the results in a tables or figures.
  5. The conclusions seems to be rather long and containing discussion. It should be modified.

Author Response

Reviewer 3

Rev.

In the current manuscript, 72% of plagiarism were detected through the manuscript by using Turnitin.

Auth.

We also received a high degree of similarity (67%) by submitting our revised manuscript to (www.turnitin.com), but we disagree with that assessment because Turnitin monitors manuscripts that have been sent previously to a Journal. Although they were not accepted, those documents remain in the corresponding university server and Turnitin database. Apparently, this is why in source no. 1 of the originality report appears "submitted to Universidad Autónoma de Nuevo León" with 62%.

I kindly ask you to consider the originality report made by iThenticate (https://app.ithenticate.com), which shows only 31% similarity.

In both Turnitin and iThenticate, much of the similarity is found to articles from our authorship, where the description of the phytochemical and biological evaluation part is very similar since the same experimental techniques are used.

In many cases, a similarity with isolated words or short fragments in some paragraphs is marked where technical standard expressions from biology, chemistry, and medicine area are used. In our opinion, there is no alternative way to express things, e.g., the names of the isolated flavonoids.

We have followed your kind advice to reformulate all sections of the manuscript to minimize these similarities and, in any case, make sure to give the corresponding credit to the source, where appropriate.

Rev.

All the novel findings should be highlighted in the abstract.

Auth.

In lines 29/30 of the revised manuscript, we added: "Amebicide activity of sakuranetin and cirsimaritin is reported for the first time in this study".

Rev.

The introduction should avoid secret the apart of result of the study (final paragraph, you tell the important result of your paper). This paragraph should mention the aims of the study, and the outline of the work

Auth.

We reformulated this paragraph in lines 57-69 of the revised manuscript.

Rev.

The data of bioactivity of all fractions, sub-fractions should be shown in the results in a tables or figures.

Auth.

We inserted a new figure on page 2 of the revised manuscript with a general scheme for the bioguided isolation.

Rev.

The conclusions seem to be rather long and containing discussion. It should be modified.

Auth.

Discussion and conclusions were reformulated in the revised manuscript. In the discussion, only references related to general antiprotozoal activity of the isolated flavonoids are now mentioned. Thus, the bibliography was reduced from 94 to 78. Conclusions were also reduced from 403 to 166 words.

Round 2

Reviewer 1 Report

Authors modify manuscript in an appropriate way. This manuscript now suitable for publication.

Reviewer 3 Report

The revised version has been approved so I have not any other comments. The authors please check the section of "references" (1,5,6,8,10 ---------) where appeared lots of mistype especially the "capital" & "scientific name".     

This manuscript is a resubmission of an earlier submission. The following is a list of the peer review reports and author responses from that submission.

Round 1

Reviewer 1 Report

The manuscript entitled “Antiprotozoal Activity Against Entamoeba histolytica of Flavonoids Isolated From Lippia graveolens Kunth”, deals with the the isolation and structure elucidation of antiamoebic compounds occurring in Lippia graveolens. Therefore, in my opinion, this manuscript should be accepted for publication with major revisions. Some point should be clarified:

General remarks

The English in the manuscript should be improved and overall it’s an extensive document, it should be resumed in some sections such as “Materials and Methods”, where some descriptive parts of the separation and purification of the compounds could be put in a scheme or in a table. Also, the document has too many references (106).

Introduction

In line 54, substitute “Newly” by another word like “Recently”.

Material and methods

Line 197 – The plants were collected in March 2011, almost 10 years. How they were stored during this period? Did you evaluate the possibility that some compounds could be degraded even in good conditions of storing?

Line 308 – Change “In vitro” to “In vitro”.

Reviewer 2 Report

Authors describes the anyiprotozoal activity of Lippia graveolens. In particular this worh was the secondo part of a previous study published in Molecules. 2014 Dec 15;19(12):21044-65. doi: 10.3390/molecules191221044. by the saem reasearch group.

The applied methodology is correct however I believe that this topic is not new considering that the identified flavonoids are well known in literature for the activity against Entamoeba histolytica please see 

Flavonoids as a Natural Treatment Against Entamoeba histolytica.

Martínez-Castillo M, Pacheco-Yepez J, Flores-Huerta N, Guzmán-Téllez P, Jarillo-Luna RA, Cárdenas-Jaramillo LM, Campos-Rodríguez R, Shibayama M.

Front Cell Infect Microbiol. 2018 Jun 22;8:209. doi: 10.3389/fcimb.2018.00209. 

Moreover the data of identification  are not required since as previously reported by many other publication. 

Extrac was obtained by maceration with methanol that is toxic for humans. So how author suggest possible application? Why they not use ethanol or hydroalcolic solution?

Conclusion required a serious revision since did not highlight the novelity ad utility of this work.

Reviewer 3 Report

Submitted for review article entitled ,, Antiprotozoal activity against Entamoeba histolytica of flavonoids isolated from Lippia graveolens Kunth” is an original paper. The authors try to show antiprotozoal activity against Entamoeba histolytica of flavonoids isolated from Lippia graveolens Kunth. This is an interesting topic because of the growing incidence of parasitic diseases all over the world.  Abstract and Introduction are reasonable clear. The methodology is good performed and reasonably clear in chromatographic section because in antiprotozoal ……is should be improved. There is lack of any antiprotozoal diagrams. Additionally , I can't find any news outstanding in this work, because even isolated, pure compounds are already described and tested (e.g naringenin).  I would suggest that the authors do at least an MTT test to show the effect of these compounds on cells in vitro.The interpretation of the results with discussion presented by authors is weak because of the small amount of presented data. In my opinion the authors should enhance your studies by different biological activites. Authors should polish up their English and correct some tiny grammar mistakes in manuscript.